# Green Synthesis and Characterization of Silver Nanoparticles Using *Artemisia terrae-albae* Extracts and Evaluation of Their Cytogenotoxic Effects

**DOI:** 10.3390/ijms26157499

**Published:** 2025-08-03

**Authors:** Moldyr Dyusebaeva, Dmitriy Berillo, Zhansaya Yesbussinova, Nailya Ibragimova, Daniil Shepilov, Sandugash Sydykbayeva, Almagul Almabekova, Nurzhan Chinibayeva, Adewale Olufunsho Adeloye, Gulzat Berganayeva

**Affiliations:** 1Faculty of Chemistry and Chemical Technology, Al-Farabi Kazakh National University, Almaty 050040/A15E3B4, Kazakhstan; moldyr.dyusebaeva@kaznu.edu.kz (M.D.); zh_yesbussinova@mail.ru (Z.Y.); shepilov2002@gmail.com (D.S.); adeloye.adewale@kaznu.kz (A.O.A.); 2Center of Agro Competence, M. Kozybayev North-Kazakhstan University, Pushkin 86, Petropavlovsk 150000/T01P7Y3, Kazakhstan; 3Faculty of Engineering and Information Technologies, Kazakh-German University (DKU), Almaty 050010/A26C7F8, Kazakhstan; nailya.73@mail.ru; 4Faculty of Natural and Technical Sciences, Zhetysu University named after Ilyas Zhansugurov, Taldykorgan 040009, Kazakhstan; sandugash78@mail.ru; 5School of Pharmacy, Department of Chemistry, Asfendiyarov Kazakh National Medical University, Almaty 050012/A35B8H9, Kazakhstan; almabekova.a@kaznmu.kz; 6Faculty of Natural Sciences and Geography, Department of Chemistry, Abai Kazakh National Pedagogical University, Almaty 050012/A05H1Y7, Kazakhstan; chinibayeva@mail.ru

**Keywords:** *Artemisia terrae-albae*, natural products, silver nanoparticles, cytogenotoxicity, ultraviolet–visible spectroscopy, transmission electron microscopy, scanning electron microscopy

## Abstract

The development of non-toxic silver nanoparticles (AgNPs) for medical and other diverse applications is steadily increasing. However, this study specifically aims to determine the cytotoxic effects of AgNPs synthesized via a green chemistry approach using aqueous-ethanol and ethyl acetate extracts of *Artemisia terrae-albae*. The photophysical, morphological, and size distribution characteristics of the synthesized AgNPs are analyzed using UV-Vis spectroscopy and transmission electron microscopy (TEM). A modified *Allium cepa* assay is employed to evaluate biological responses, including root growth, root number, and mitotic index. In this assay, the cell cycles of onion bulbs are synchronized and pre-incubated at 4 °C for 72 h prior to treatment. This study reveals that the AgNPs synthesized from the ethanol extract exhibit notable stability and higher cytotoxicity activity, with a root length of 0.6 ± 0.13 cm, root number of 16 ± 6.88, and mitotic index of 25.0 ± 2.6. These values are significantly more cytogenotoxic than those observed for the ethyl-acetate-derived nanoparticles, which show a root length of 0.8 ± 0.17 cm, root number of 18 ± 6.27, and mitotic index of 36 ± 3.6. These findings highlight the potential of green-synthesized AgNPs as effective cytotoxic agents, especially those obtained from ethanol extract, possibly due to a greater influence of the quantity of diverse phenolic compounds present in the complex mixtures than in the ethyl acetate extract, which otherwise enhanced their morphology, shape, and size. These, overall, contributed to the biological activity.

## 1. Introduction

Utilization of medicinal plant extracts as the primary source in the application of the green synthesis method for silver nanoparticles has gained substantial interest in diverse fields of medical applications due to its environmental friendliness, sustainability, energy efficiency, and viability as an economic approach. These are advantages over conventional physical and chemical synthesis methods, which involve the use of biogenic and other hazardous reagents that increase the generation of toxic-waste by-products. Various parts of plants—such as leaves, stems, bark, seeds, roots, and flowers—have been employed in green synthesis methods, as they function as both reducing and stabilizing agents, primarily due to the presence of various bioactive phytochemicals, such as flavonoids, terpenoids, and phenolic acids.

In recent times, the green synthesis method has been used extensively by various research groups in the preparation of AgNPs, while their photophysical and morphological characteristics have been determined. For instance, *Psidium guajava* leaf extract yielded silver nanoparticles with an average size of 14 nm in diameter, spherical morphology, polydispersity index of 0.43, and −15.6 mV zeta potential, leading to satisfactory colloidal stability. In fact, it has been established that the average diameter size for such nanoparticles is between 1 and 200 nm [1]. A comprehensive characterization of green-synthesized AgNPs is imperative for elucidating their physicochemical attributes and assessing their functional capabilities. Analytical techniques employed include UV–visible spectroscopy, Fourier transform infrared spectroscopy, X-ray diffraction, transmission and scanning electron microscopy (TEM and SEM), dynamic light scattering (DLS), energy-dispersive X-ray spectroscopy, and zeta potential measurements [2,3].

Several silver nanoparticles prepared by green synthesis have been investigated for their biological activities. In studies reported by Manzoor and co-workers [4], AgNPs were prepared using *Trillium govanianum* rhizome extract. The antioxidant and anti-inflammatory activities of the nanoparticles were determined and found to be highly comparable to the standard drugs ascorbic acid and diclofenac sodium, respectively. A similar report on aqueous leaf extracts of *Azadirachta indica* and *Ocimum sanctum* exhibited dose-dependent antioxidant activity, which was also comparable to standard ascorbic acid. In fact, the microbiological activity of the AgNPs was found to suppress the growth of *Enterococcus faecalis*, as well as reduce biofilm formation [5]. Using a *Trichoderma harzianum* fungal extract, Korcan et al. prepared and reported AgNPs showing high radical scavenging activity and anti-inflammatory and antibacterial properties, as well as potential as an inhibitor of protein denaturation [6].

In the areas of anticancer research, AgNPs prepared from plants, such as *Trillium govanianum*, *Azadirachta indica*, Aerva lanata, *Ocimum sanctum*, Sargassum wightii, Acorus calamus, and Artemisia species, namely, A. abrotanum, A. absinthium, A. annua, A. dracunculus, and A. vulgaris, have been found to exhibit a broad range of dose-dependent cytotoxic and genotoxic properties against MCF-7 breast cancer cells and V79 Chinese hamster lung cells. It has been reported that phytogenic AgNPs exhibited selective cytotoxicity, exerting minimal toxicity toward normal cells while inducing substantial cytotoxic effects in malignant cells, thereby underscoring their therapeutic potential in oncology and targeted drug delivery applications [7,8,9].

Despite the promising biomedical applications, the inherent variability and complexity of phytochemical constituents present significant challenges for the standardization and reproducibility of green synthesis protocols. Moreover, the toxicological profile of AgNPs—particularly in relation to long-term exposure and biodistribution—remains insufficiently characterized and requires further systematic investigation. This study aimed to analyze the effects of AgNPs on biological systems, with a specific focus on their cytogenotoxic properties. Cytotoxic effects are studied for their ability to signal damage to fundamental building blocks of life, such as cells and genetic material. They play an important role in drug development and environmental monitoring and protection, as well as in research on the cell cycle and DNA repair mechanisms. Applications have also been identified in fields such as human health, disease prevention, birth defect research, and the study of genetic disorders [10,11,12,13,14,15,16,17].

In this work, silver nanoparticles were synthesized via a green method using aqueous-ethanol and ethyl acetate extracts of *Artemisia* spp. The selection of *Artemisia terrae-albae* in our study was based on several key factors. First, this plant possesses a rich phytochemical composition (flavonoids, phenolic compounds, terpenoids, phenolic acid, glycosides, etc.), which enables it to serve a dual role in nanoparticle synthesis, both as a reducing agent and as a stabilizer (capping agent), eliminating the need for synthetic reagents [18]. Second, *A. terrae-albae* is an endemic species native to the arid regions of Kazakhstan and Central Asia. Its traditional use in folk medicine makes it a promising candidate for the development of localized green technologies.

Moreover, the use of organic solvents (ethanol and ethyl acetate) in this study enables the selective extraction of different groups of bioactive compounds, allowing for controlled synthesis of nanoparticles with desired properties. We do not position our approach in opposition to classical methods, but rather propose it as an alternative route—particularly relevant for the development of functionalized AgNPs with tailored properties for biomedical applications.

Although the cytogenotoxic properties have previously been analyzed using different methods—including in situ hybridization, karyotyping, SNP arrays, comparative genomic hybridization, array-based CGH, next-generation sequencing, and quantitative PCR [19,20,21,22,23,24]—this assay is particularly useful for analyzing microscopic parameters such as the mitotic index (MI), chromosomal aberrations (CA), and micronuclei formation [25,26,27,28,29,30,31].

Herein, we report the assessment of AgNP cytotoxicity via modification of a known method, the *Allium cepa* assay, which involved pre-treating bulbs at 4 °C for 72 h to synchronize the mitotic cycle in root meristem cells.

## 2. Results and Discussion

### 2.1. UV-Vis Absorbance Spectral Studies of AgNPs@Artemisia EtOH Diluted in Water

A previous investigation focused on isolating and characterizing the bioactive compounds present in *Artemisia cina* Berg raw material. Key constituents were identified via chromatography–mass spectrometry. The antiviral efficacy of several extracts was then evaluated using a modified limiting-dilution assay (Reed–Muench method). Notably, the first confirmation of *Artemisia cina* Berg extracts exhibiting activity against the SARS-CoV-2 virus was reported recently [32]. We designed a simple method of AgNP preparation using *Artemisia* EtOH/H_2_O extract under mild conditions (Figure 1).

The surface plasmon resonance (SPR) peak of AgNPs@ArtEtOH/H_2_O at 420 nm confirmed the spontaneous formation of AgNPs (Figure 2A–D). The use of the *Artemisia* extract that is diluted 40 times in water is preferable to using the one diluted 60 times in water, perhaps because it contains a higher concentration of compounds valuable for AgNP stabilization at the seeding stage. In order to check that all silver phytocomplexes had been reduced to AgNPs, the reaction mixture was additionally treated in an ultrasonic bath for 2 min under standard conditions. The SPR peak of AgNPs after additional treatment did not change, which indicates a complete reduction. A control experiment was carried out to test for the presence of silver ions in solution after AgNP formation. The reaction with sodium chloride did not result in the formation of a white precipitate of silver chloride. Moreover, the addition of a glucose solution did not trigger an increase in the SPR peak of AgNPs, again indicating the absence of free silver ions. However, a discussion of the mechanism of AgNP stabilization by plant extract and the influence of various trigger parameters (UV or microwave irradiation, pH, temperature) on the induction of AgNPs was reported earlier [33]. For instance, during the formation of AgNPs, phenolic groups of flavonoids participate in the reduction of Ag ions, which, in turn, are oxidized to quinone functional groups [34]. This suggests that, in addition to the aforementioned, other functional groups, such as the aldehydic groups in vanillin or other aromatic aldehydes, may participate in the process of reduction, as well as the provision of subsequent stabilization of the AgNPs through carboxyl groups, and, finally, the formation of chelates with oxygen- and nitrogen-containing functional groups.

The red line is attributed to the absorption spectrum of the AgNPs@Art EtOAc ×60 diluted plant extract. Initially, 1 part dry *Artemisia* EtOAc extract was diluted in 60 parts water, followed by the synthesis of silver nanoparticles. The resulting suspension was highly concentrated for direct UV-vis detection and was further diluted 10-fold before measurement (Figure 2C). The blue line corresponds to the absorption spectrum of AgNPs@Art EtOAc ×600 ×2: 1 part EtOAc extract was first diluted in 60 parts water, AgNP synthesis was carried out, and the resulting solution was diluted 10 times and then additionally 2 times with water before UV-vis analysis. The green line is attributed to the initial absorption spectrum of *Artemisia* EtOAc extract diluted in 60 parts water, without nanoparticle synthesis.

### 2.2. TEM Surface Morphological Studies of Artemisia EtOH/H_2_O and Artemisia EtOH/H_2_O/EtAc

In Figure 3, a series of TEM images (transmission electron microscopy) of silver nanoparticles is presented. The images are labeled A–H and represent different magnifications ranging from ×40,000 to ×200,000. The samples are divided into two groups: (A–D) (AgNPs@*Artemisia* EtOH/H_2_O/EtAc extract diluted with water ×60) and (E–H) (AgNPs@*Artemisia* EtOH/H_2_O/EtAc extract diluted with water ×45). One can observe quite a few large particles, with sizes exceeding 100 and 150 nm. These particles have a tendency to precipitate within a short period of time and therefore are less likely to be detected using light scattering methods.

In the TEM images, silver nanoparticles exhibit various shapes, such as spherical, ellipsoidal, polyhedral, and irregular forms. In the AgNPs@*Artemisia* EtOH extract diluted with water ×60 group (images A–D), the particles are predominantly round or slightly ellipsoidal. However, some aggregated structures with irregular shapes are also observed. The size distribution of the particles in this group appears more uniform, and they are generally well-dispersed, with a moderate degree of aggregation [35].

In addition, in the AgNPs@*Artemisia* EtOAc extract diluted with water ×45 (Figure 3E–H), the particle shapes are more diverse. Spherical particles and distinct polyhedral nanoparticles, including triangular and hexagonal structures, can be observed. This group exhibits a higher degree of aggregation, which may be associated with differences in the synthesis or stabilization of the nanoparticles. In certain areas, dense clusters can be seen, where particles are in direct contact, forming complex clustered structures.

At magnifications of ×120,000–200,000, it becomes evident that some particles have smooth edges, while others display well-defined facets characteristic of crystalline nanoparticles. In some cases, core–shell nanoparticles can be distinguished, suggesting compositional heterogeneity. Meanwhile, both groups present rounded particles that predominate, but the AgNPs@*Artemisia* EtOAc extract ×45 group contains structures with more distinct crystalline geometry and a stronger tendency to form aggregates.

### 2.3. Surface Particle Distribution and Size Analysis of AgNPs@Artemisia EtOH/H_2_O and AgNPs@Artemisia EtOAc/H_2_O

The size distribution of AgNPs is presented in histograms (Figure 4), obtained from the analysis of TEM images using ImageJ software, version 1.53t. Although it is well known that nanoparticles prepared from natural plant materials are eco-friendly, it is very challenging to obtain highly monodispersed particles due to the complex nature of the mixtures of phytochemicals present therein, unless the synthesis conditions are carefully optimized [36]. Histogram A demonstrates a broad distribution of nanoparticle sizes, with the highest frequency in the 18–30 nm range (Figure 4A), followed by a gradual decrease in the number of particles. The distribution is skewed to the right, which possibly indicates the presence of larger particles. Histogram 4B is sparser and biased towards larger particles. Both distributions are asymmetrical. Most likely, the extract has fewer compounds that have the ability to rapidly stabilize growing AgNPs, which leads to the growth of large particles. The larger the particle, the smaller the ratio of surface to size, and in turn, the less activity. Overall, Histogram A indicates a more uniform particle distribution, suggesting a stable synthesis process, whereas Histogram B points to a less uniform distribution or an insufficient sample representation. In both cases, most particles are smaller than 50 nm, confirming their nanometer scale.

### 2.4. Cytotoxicity Studies of AgNPs@Artemisia EtOH/H_2_O Extract and AgNPs@Artemisia EtOH/EtOAc Extract

Recently, we utilized a solvent–solvent extraction method. *A. terrae-albae* was selected for its exceptionally high α-santonin content, and solvent extraction was employed to efficiently isolate its bioactive constituents. In Extract 3, α-santonin constituted 66.33% of the yield (RT 18.2 min), alongside minor amounts of 2,3-dihydro-4H-pyran-4-one (3.2%), 3,5,5-trimethylcyclohexylisophosphofloride (6.73%), lumisantonine (3.74%), and 6-nitro-2-phenyl-4-quinolinol (8.0%). Extract 4 contained 39.39% α-santonin, 10.05% butanoic acid, and 3.28% anhydro-β-retinol, while Extract 7 mirrored Extract 3, with 66.35% α-santonin, plus 4.9% 4-H-pyran-4-one-2,3, 5.9% 7-ethyl-4-nonanone, and 2.58% lumisantonine. Since antiviral activity was ranked as No. 4 > No. 3 = No. 7 and tracked with α-santonin levels, these results implicate santonin as the primary active component [32].

*Artemisia vulgaris* essential oil (EO) is primarily composed of monoterpenes (64.5%) and sesquiterpenes (12.55%), with (Z)-sabinol, trans-sabinyl acetate, eucalyptol, and (E)-piperitol as the major components, collectively accounting for 45.63% of the EO. The oil demonstrated strong phytotoxic effects, significantly inhibiting seedling root growth of both monocots (*Setaria viridis* and *Poa annua*) and dicots (*Amaranthus retroflexus* and *Medicago sativa*) in a dose-dependent manner. At 5 mg/mL, root development was suppressed by over 96% in all tested species. Additionally, the EO exhibited strong insecticidal activity against *Aphis gossypii* and antimicrobial effects, particularly against *E. coli*, with moderate activity against *A. niger*, *V. dahliae*, and *B. subtilis* [37]. *Artemisia*-derived products have shown potential in protecting crops against a wide range of pests and pathogens, including fungi, bacteria, insects, nematodes, and weeds. Most research has focused on plant extracts—particularly essential oils—rather than on isolated compounds, although terpenoids, flavones, coumarins, and phenolic acids have been identified as key active components. Extracts often outperform isolated compounds, likely due to synergistic effects, which may also reduce the risk of resistance development. However, concerns remain about their toxicity to non-target organisms, and there is limited research on their environmental impact and field-level effectiveness [38].

Previously, the toxicity of an aqueous extract of *Artemisia afra* was tested in vivo. Intraperitoneal injections (1.5–5.5 g/kg) showed a dose-dependent increase in both mortality and adverse behavioral effects. In contrast, oral administration (2–24 g/kg) led to behavioral and mortality effects that were not clearly dose-dependent. The estimated LD50 values were 2.45 g/kg for intraperitoneal and 8.96 g/kg for oral administration, indicating higher toxicity through injection than ingestion [39]. An inexpensive approach to toxicity evaluation can be applied. The biological impact of *Artemisia annua* L. on the small white butterfly (*Pieris rapae* L.) was investigated under controlled laboratory conditions (16:8 h light–dark cycle at 25 ± 1 °C and 65 ± 5% relative humidity). The study examined mortality, growth, feeding behavior, and biochemical responses. The LC_50_ and LC_25_ values for *A. annua* were determined to be 9.4% and 3.6%, respectively. Even at the lowest tested concentration (0.625%), the extract showed a deterrent effect of 29.8% [40]. The effects of *Artemisia annua* extracts from Brazil and China were studied on normal leukocytes and Molt-4 leukemia cells, with dihydroartemisinin (DHA) used as a reference. Cell viability was assessed over 72 h at varying artemisinin concentrations, and antioxidant activity was measured using the ORAC assay. Both extracts showed strong antioxidant properties and selective toxicity toward Molt-4 cells. DHA was more effective in killing cancer cells than either extract at later time points, while all treatments were less toxic to normal leukocytes, with DHA showing the highest safety margin [41]. *Artemisia annua* extract demonstrated a strong inhibitory effect on AGS gastric cancer cells, primarily through the induction of apoptosis, while showing lower toxicity toward normal L929 cells. Among the different extracts, the methanol extract exhibited the most potent growth-inhibitory activity (IC50: 500 µg/mL) and induced apoptosis more effectively than others. These findings suggest that *A. annua* may have promising potential for use in the prevention or treatment of gastric cancer [42].

In order to decrease the use of animals, we employed an alternative method, resulting in faster data collection.

The rate of root growth is a very sensitive indicator of the cytotoxicity of a substance and correlates with microscopic findings. The characteristics of the root system of *Allium cepa* after 72 h of aging in colloidal silver solutions are presented in Table 1.

As depicted in Table 1, the average length of roots cultivated in colloidal solutions of silver nanoparticles was significantly reduced compared to the control, showing a 4- to 6-fold decrease. Moreover, the number of roots in individual bulbs was higher under the influence of the AgNPs@*ArtemisiaEtOH/*EtOAc extract group. This indicates a substantial inhibition of root growth. Moreover, cytotoxicity was evident in the visual appearance of the roots. Upon visual assessment, notable differences between the experimental variants and the control were observed. The roots grown in colloidal silver solutions exhibited morphological deformities, curvature, reduced length, and slight thickening, possibly attributable to impaired proliferation of mitotic cells in the meristem and a decrease in the mitotic index. Additionally, a change in root color was noted, with the tips appearing dark brown at the meristem level.

Table 2 presents the mitotic cycle indices and meristematic index in meristematic cells of *Allium cepa* roots cultured in the studied silver nanoparticle solutions.

Among all tested groups, the control group had the highest mitotic index, averaging 64.4 ± 6.4%. In contrast, a significant decrease in the mitotic index to 25.0 ± 2.6% was observed after exposure to AgNPs synthesized using *Artemisia* EtOH/H_2_O extract diluted 40-fold with water, indicating strong inhibition of cell division. Moderate mitotic suppression was recorded for nanoparticles obtained using the EtOAc extract diluted 45-fold with water, with a mitotic index of 35.8 ± 3.6% (Table 3).

As expected, all mitotic phases (prophase, metaphase, anaphase, and telophase) were present in the control group, with prophase dominating and only minimal abnormalities observed. In the EtOH extract group, cells were predominantly arrested in prophase; only a few reached metaphase or anaphase, and no telophase cells were observed, suggesting mitotic arrest at early stages.

In the EtOAc extract group, prophase also predominated, but some cells reached metaphase and anaphase, and a small number progressed to telophase. This indicates a slightly less toxic effect compared to AgNPs synthesized from the EtOH extract.

These observations correlate with the particle size distribution: although nanoparticles smaller than 50 nm predominated in both groups (Figure 4), the TEM histograms showed a broader and more heterogeneous distribution in the EtOAc extract group, including a higher proportion of larger particles (Figure 4B). This may explain the lower cytotoxicity, as larger nanoparticles generally exhibit lower reactivity due to a reduced surface-to-volume ratio.

### 2.5. Cell Mitosis Studies of AgNPs@Artemisia EtOH/H_2_O Extract and AgNPs@Artemisia EtOH/EtOAc Extract

The study by Cetkovic et al. examined the genotoxic and cytotoxic effects of a commercially available *Artemisia annua* L. tincture, widely used due to its artemisinin content for malaria, cancer, and antiviral purposes. Using comet and neutral red uptake assays, the tincture showed dose-dependent DNA damage and cytotoxicity in both peripheral blood and cell cultures. Significant DNA damage and reduced cell viability were observed at lower dilutions (higher concentrations) of the extract. These findings suggest that *A. annua* tincture should be used with caution, particularly when not adequately diluted [43].

Previously, extracts of *Artemisia vulgaris* and *Artemisia alba* were evaluated for their genotoxic and cytotoxic effects. Both extracts increased the micronucleus (MN) frequency in peripheral blood lymphocytes at most tested concentrations, with *A. alba* also significantly affecting the nuclear division index (NDI) across all doses. When used alongside the genotoxic agent MMC, both extracts reduced the MN frequency and NDI in a dose-dependent manner. *A. alba* showed strong cytotoxicity in SW-480 cancer cells, while *A. vulgaris* exhibited cytotoxic effects only in combination with MMC after prolonged exposure; importantly, neither extract harmed human periodontal ligament stem cells [44]. High concentrations (375 and 500 μg/mL) of *Artemisia herba-alba* extract significantly suppressed cell division in mouse bone marrow cells and induced genotoxic effects such as sister chromatid exchanges and micronuclei formation. The extract also reduced cell viability in bone marrow cultures in a dose-dependent manner. IC50 values ranged from around 484 to 513 μg/mL across 24, 48, and 72 h of exposure. These results indicate that *A. herba-alba* possesses notable genotoxic and cytotoxic effects at elevated concentrations [45].

The study results demonstrate growth inhibition at both concentrations of *AgNPs@Artemisia* EtOH extract/H_2_O *and AgNPs@Artemisia* EtOH extract/EtOAc compared to the control group. AgNPs led to a significant reduction in cell division in the meristem tissue cells of *Allium cepa*. While cell mitosis was observed in the control group, samples containing nanoparticles showed almost no cell division (Figure 5A–D).

### 2.6. Cytogenotoxicity Studies of AgNPs@Artemisia EtOH extract/H2O and AgNPs@Artemisia EtOH extract/EtOAc

As presented below, Figure 6 shows the identified aberrations in *Allium cepa* cells during incubation in solutions with silver nanoparticles. It was observed that the smaller the particle diameter, the greater the modification of the MI compared to the control group. Stray chromosomes (lagging chromosomes) are formed due to abnormalities of the spindle. Chromosome adhesion occurs for several reasons, such as translocation, inversion of a chromosome segment, chromatid entanglement, and disruption of the activity of enzymes such as DNA topoisomerase II, which leads to non-separation of chromosomes in anaphase. Cell polyploidy is due to the occurrence of C-mitosis [46]. Bridges and fragments are formed by chromatid breaks, which lead to the formation of dicentric chromosomes that simultaneously pull on both poles in the anaphase stage [47]. Bridges can also form due to breakdowns, fusion of chromosomes and chromatids, and changes in the activation of replication enzymes [48]. The micronucleus consists of small fragments of chromatin that are formed as a result of chromosomal breakage, or of entire chromosomes that do not migrate during anaphase as a result of spindle dysfunction [47,49]. Cohesion (chromosomal adhesion) occurs because of chromosome condensation, which is due to DNA degradation/depolymerization, partial dissolution of nucleoproteins, breakage, and chromatid exchanges. Chromosomal adhesion indicates a highly toxic, irreversible effect, likely leading to cell death [50,51].

The occurrence of various types of aberrations in the chromosomes of meristematic cells of *A. cepa* roots can be associated with the clastogenic and aneugenic influence of the studied colloidal solutions of silver nanoparticles due to the formation of free radicals that cause irreversible damage in the processes of replication, repair, recombination, and transcription of DNA and, as a result, the observed necrobiotic changes in cells.

Changes in root growth are recorded, and cell analysis is performed using microscopy [52]. It has been established that the outcomes derived from onion studies exhibit a high correlation with results obtained in investigations involving mammalian cells, including those of humans [53].

*AgNPs@Artemisia* EtOH extract suspension exposure significantly reduces the mitotic index, with the 40 nm nanoparticles causing the strongest inhibition of cell division. Larger nanoparticles (73 nm) exhibit a slightly milder effect, allowing a few cells to progress beyond metaphase. Potential toxicity mechanism: Smaller AgNPs likely penetrate cells more efficiently, disrupting mitotic progression and leading to cell cycle arrest. AgNPs cause mitotic arrest at prophase, with smaller nanoparticles (40 nm) exerting a stronger inhibitory effect than larger ones (73 nm). The complete absence of anaphase and telophase in the 40 nm AgNP group suggests severe disruption of cell division. The 73 nm AgNPs also inhibit mitosis but allow a small fraction of cells to progress beyond metaphase. The observed abnormal mitotic figures thus indicate possible genotoxic effects.

## 3. Materials and Methods

### 3.1. Reagents

Ethyl acetate 99% and Ammonia solution 10% were purchased from Sigma-Aldrich (Saint Louis, MO, USA); Nitric acid silver (AgNO_3_ ≥ 99.9999%) was acquired from Labrpharma (Almaty, Kazakhstan); Ethanol 96% was produced by Talgar ethanol factory (Talgar, Kazakhstan); and 0.1 N Hydrochloric acid was used.

### 3.2. AgNP Synthesis

In the first experiment, 6.6 g of ethanolic extract of *Artemisia terrae-albae* Krasch. (*Asteraceae* L.) was dissolved in water (1:40) and then combined with 13.2 g of 1.5 mM AgNO_3_ aqueous solution. The initial pH of the reaction mixture, at 4.6, was adjusted to 9.8 with the addition of 10% ammonia solution in a few drops. A dark solution was observed immediately with the adjustment of the pH value, leading to an SPR peak at 445 nm. Subsequent experiments involve the addition of 0.061 g of the ethyl acetate extract of *Artemisia terrae-albae* to 2.75 mL of 8% solution of ethyl acetate in water. The resulting plant extract solution was diluted 45 times before adding it to 5.5 mL of 1.5 mM AgNO_3_ aqueous solution. The pH was again adjusted to 9.4 using 10% ammonia solution, and a two-phase mixture was obtained. A color change to dark gray was observed after 30 min of incubation at room temperature in a dark place. The solution further underwent a 2-min treatment in an ultrasonic bath, and UV-Vis was recorded (Figure 1) [54].

### 3.3. TEM Analysis of AgNPs

The morphology and size of AgNPs*@Artemisia* EtOH extract *and AgNPs@Artemisia* EtOAc extract stabilized by the plant extract were examined using a JEOL JEM-1400 Plus transmission electron microscope. The sample was prepared by dispersing AgNPs onto a carbon-coated copper grid (mesh size: 200) and allowing it to dry. Images were captured under an electron acceleration voltage of 80 kV. The obtained TEM images were analyzed using ImageJ software, where area measurements were performed to determine the average particle diameter.

### 3.4. Methodology for Determining Chromosomal Aberrations

To conduct the analysis, 5 (five) onion bulbs from *Allium cepa* L. (*Alliaceae*) were used for each sample. The bulbs were preconditioned at room temperature in distilled water for 24 h to assess their germination potential. Subsequently, the bulbs were placed in the test colloidal solutions of silver nanoparticles (10 mL) for 72 h. Control bulbs were germinated in distilled water at a temperature of +25 (±1) °C for the same 72-h period. It is crucial to monitor the levels of the test solutions and ensure direct contact between the onion root system and the solutions, according to a previously published protocol [55]. After the incubation period, the root tips of the bulbs were excised and transferred into vials containing a modified Carnoy’s fixative (a mixture of 3 parts 96% ethanol and 1 part glacial acetic acid). After 2 h, the roots were rinsed in 70% ethanol and stored in a refrigerator at +6 °C.

The following parameters were evaluated during the experiment:The average number of roots;Root length;The distribution of cells in the root tip of *Allium cepa*.

Before conducting microscopic analysis, the excised root tips were placed in a 0.1 N solution of hydrochloric acid at room temperature for 5 min to pre-hydrolyze the cell walls. The apical portions of the roots were then stained with a 2.5% orcein solution, and *squash preparations* were made—a technique in which the stained tissue is gently squashed between a microscope slide and a cover slip to produce a monolayer suitable for observing mitosis and chromosomal aberrations. For cytogenetic analysis, two roots from each bulb were examined under a binocular microscope (LEICA DM-1000, Leica Microsystems GmbH, Wetzlar, Germany) at magnifications of ×20–40.

The mitotic index (MI %) was calculated, representing the ratio of the number of cells undergoing mitosis to the total number of analyzed cells. Among mitotic cells, the proportion of cells in the stages of prophase (P %), metaphase (M %), anaphase (A %), and telophase (T %) was recorded.

*The mitotic index* (MI) was calculated as the proportion of all dividing cells to the total number of analyzed cells:MI, % = (P + M + A + T))/N(1)
where

MI—mitotic index;P—total number of cells in prophase;M—total number of cells in metaphase;A—total number of cells in anaphase;T—total number of cells in telophase;N—total number of analyzed cells.

*The prophase index* (PI) was defined as the ratio of cells in prophase to the total number of analyzed cells:PI, % = (P)/(P + M + A + T)(2)

The *metaphase index* (TMI) was calculated as the proportion of cells in metaphase to the total number of analyzed cells:TMI, % = (M)/(P + M + A + T)(3)

The *anaphase index* (AI) was calculated as the proportion of cells in anaphase to the total number of analyzed cells:AI, % = (A)/(P + M + A + T)(4)

The *telophase index* (TI) was defined as the ratio of cells in telophase to the total number of analyzed cells:TI, % = (T)/(P + M + A + T)(5)

### 3.5. Statistical Analysis

Statistical data processing was conducted using Microsoft Office Excel 2010 software. Statistical significance between groups was determined at *p* ≤ 0.05. Following the acquisition of numerical values from the primary data, calculations were performed to obtain the arithmetic mean (M) and the standard error of the mean (m).

## 4. Conclusions

In this work, for the first time, we have examined the influence of AgNPs synthesized by a green method using *Artemisia terrae-albae* to establish their cytogenotoxic properties by an *Allium cepa* assay. It is known that changes in the mitotic index are considered a parameter of cytotoxicity. It was found that the *AgNPs@Artemisia* EtOAc extract enhanced root growth to 0.6 ± 0.13, with an average root number of 16 ± 6.88 and a mitotic index of 25 ± 2.6, indicating a potential cytogenotoxic effect. A comparative investigation of the silver nanoparticle composite in the EtOAc extract also demonstrated a root growth of 0.8 ± 0.17, with an average root number of 18 ± 6.27 and a mitotic index of 36 ± 3.6, suggesting moderate cytotoxicity relative to the *AgNPs@Artemisia* EtOH extract composition. In conclusion, these results have shown that solvent systems play a critical role, particularly in the modulation of cytogenotoxicity of synthesized AgNPs. Subsequent investigations will be directed towards the evaluation of extracts from additional medicinal plant sources and the mitigation of cytotoxic effects through targeted modification of nanoparticle–biomolecule interactions.

## Figures and Tables

**Figure 1 ijms-26-07499-f001:**
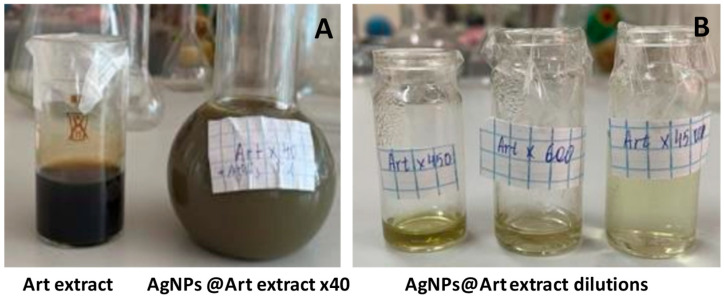
Photo of solutions of (**A**) *Artemisia* EtOH/H_2_O diluted before the synthesis and AgNPs@*Artemisia* EtOH/H_2_O ×40 dilution of initial plant extract; (**B**) AgNPs@*Artemisia* EtOH/H_2_O ×45 and AgNPs@*Artemisia* EtOH/H_2_O ×60 dilution of initial plant extract for the synthesis of particles and subsequent dilution to 10 and 100 times, respectively.

**Figure 2 ijms-26-07499-f002:**
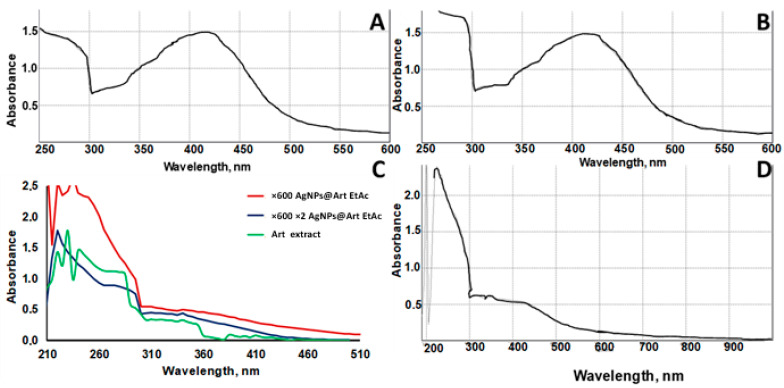
UV-Vis spectra of AgNPs stabilized by the *Artemisia* extract: (**A**) before ultrasonic treatment of AgNPs@*Artemisia* EtOH/H_2_O/EtAc diluted ×40 + AgNO_3_ and (**B**) after ultrasonic treatment of *Artemisia* EtOH/H_2_O diluted ×40; (**C**) AgNPs@*Artemisia* EtOH/H_2_O/EtAc extract diluted ×60 with water before ultrasonic treatment and then additionally diluted 10 times with water and the control sample *Artemisia* EtOH/H_2_O/EtAc extract; (**D**) *Artemisia* EtOH/H_2_O/EtAc extract diluted ×60 with water after ultrasonic treatment.

**Figure 3 ijms-26-07499-f003:**
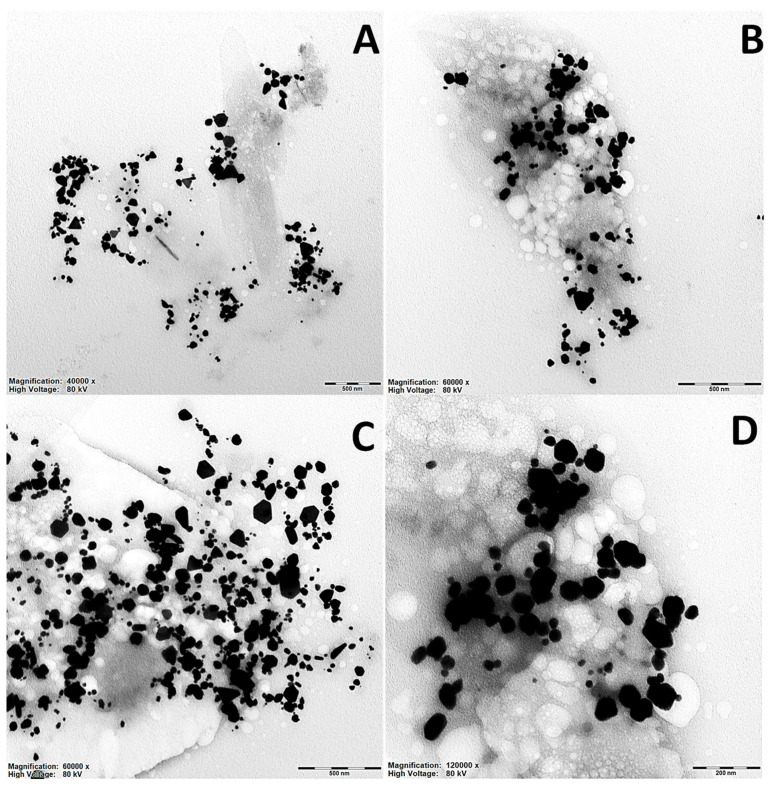
TEM images of silver nanoparticles synthesized using *Artemisia* extracts: (**A**–**D**)—AgNPs@*Artemisia* EtOH (diluted ×40 with H_2_O); (**E**–**H**)—AgNPs@*Artemisia* EtOAc (diluted ×45 with H_2_O). Magnifications and scale bars: (**A**,**E**)—×40,000, scale bar = 500 nm; (**B**,**C**)—×60,000, scale bar 500 nm; (**D**,**G**)—×120,000, scale bar 200 nm; (**H**)—×200,000, scale bar 100 nm.

**Figure 4 ijms-26-07499-f004:**
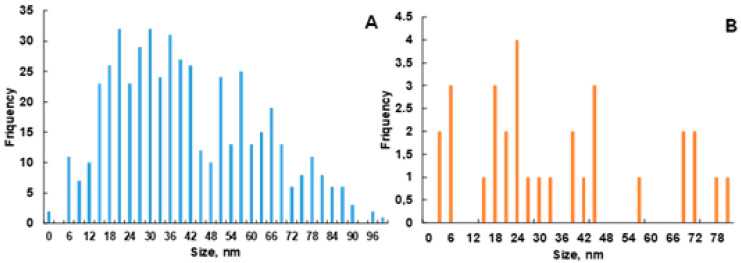
Histograms of AgNP size distribution based on TEM image analysis using ImageJ software: (**A**) AgNPs@*Artemisia* EtOH/H_2_O extract ×40; (**B**) AgNPs@*Artemisia* EtOH/EtOAc extract ×45.

**Figure 5 ijms-26-07499-f005:**
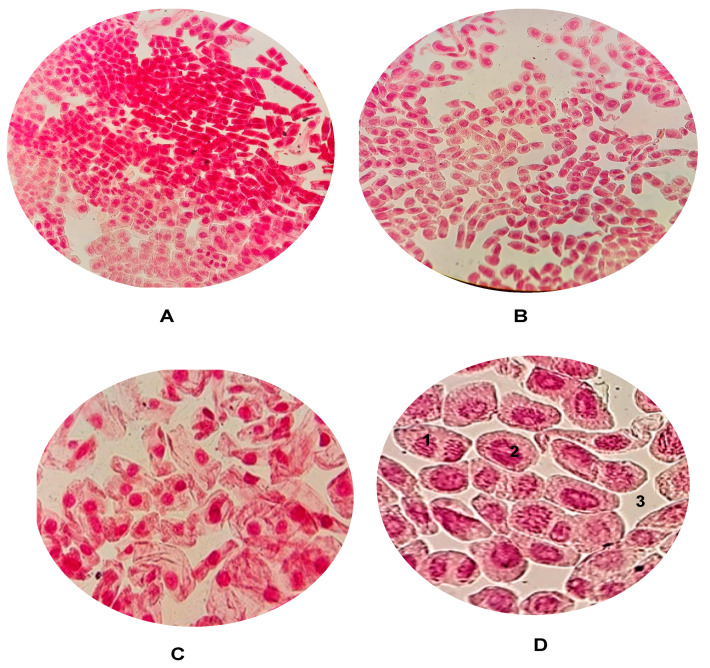
Identified aberrations in meristematic cells of *Allium cepa* roots (control—water): (**A**) cell mitosis; (**B**) lagging chromosomes; (**C**) lagging chromosomes; (**D**) cell mitosis: 1—anaphase; 2—prophase; 3—interphase.

**Figure 6 ijms-26-07499-f006:**
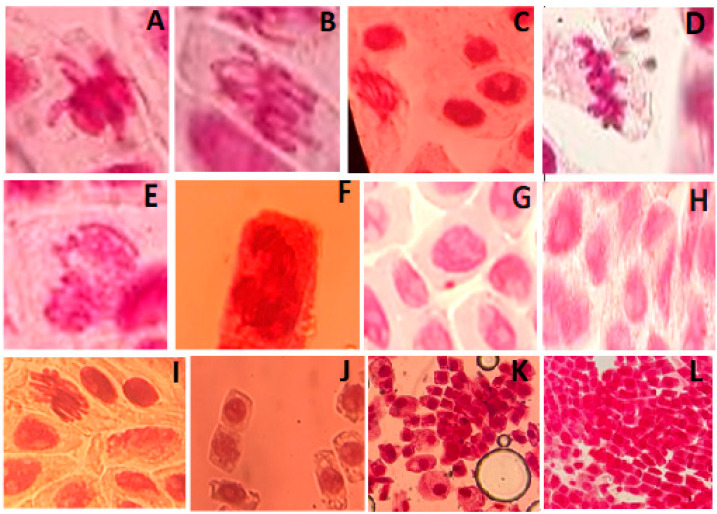
Microphotographs ×1000 of aberrations in the meristematic cells of *Allium cepa* roots during incubation in solutions of silver nanoparticles from *AgNPs@Artemisia* EtOH/H2O extract *and AgNPs@Artemisia* EtOAc extract: (**A**–**C**) lagging chromosomes; (**D**) C-mitosis; (**E**,**F**) chromosome stickiness; (**G**) micronucleus in prophase. Nuclear lesions: vacuolization; (**H**–**L**) nuclear lesions: vacuolization; necrosis; stickiness.

**Table 1 ijms-26-07499-t001:** Number and length of *Allium cepa* roots grown in the studied colloidal silver solutions at concentration: M ± m.

Experiment Number	Number of Bulbs	Number of Roots	The Length of the Roots cm	Number of Roots M ± m	The Length of the Roots, cm M ± m
Control (distilled water)					
1	5	10	1.6	15.2 ± 5.07	3.48 ± 3.11
2	5	11	1.9		
3	5	20	2.7		
4	5	21	2.2		
5	5	14	9.0		
AgNPs@*Artemisia* EtOH/H2O diluted by water ×40 (70 + 35 H_2_O)					
1	5	8	0.4	16.4 ± 6.88	0.58 ± 0.13
2	5	12	0.6		
3	5	19	0.5		
4	5	26	0.7		
5	5	17	0.7		
AgNPs@*Artemisia* EtOH/EtOAc extract diluted by water ×45 (73)					
1	5	18	0.9	17.6 ± 6.27	0.84 ± 0.17
2	5	9	0.7		
3	5	26	1.1		
4	5	15	0.7		
5	5	20	0.8		

**Table 2 ijms-26-07499-t002:** Proportion of cells in the root system of *Allium cepa* in different phases of mitosis under the influence of various colloidal solutions of silver nanoparticles.

Experiment Number	The Number of Cells Analyzed	The Number of Cells in				Mitotic Index (%)
		Prophase	Metaphase	Anaphase	Telophase	
Control (distilled water)						
1	684	664	12	6	2	68.4
2	648	638	7	3	0	64.8
3	657	642	10	4	1	65.7
4	634	623	8	3	0	53.4
5	695	681	8	5	1	69.5
M ± m	664 ± 25	650 ± 23	9 ± 2	4 ± 1.3	0.8 ± 0.8	64.4 ± 6.4
AgNPs@*Artemisia* EtOH/H_2_O diluted by water ×40 (70 + 35 H_2_O)						
1	243	243	0	0	0	24.3
2	237	237	0	0	0	23.7
3	219	218	1	0	0	21.9
4	285	283	2	0	0	28.5
5	267	266	1	0	0	26.7
M ± m	250 ± 26	249 ± 25	0.8 ± 0.8	0	0	25.0 ± 2.6
AgNPs@*Artemisia* EtOH/EtOAc extract diluted by water ×45 (73)						
1	320	320	0	0	0	32.0
2	409	404	3	1	1	40.9
3	346	343	2	1	0	34.6
4	389	388	1	0	0	38.9
5	332	331	1	0	0	33.2
M ± m	359 ± 38	357 ± 37	1.4 ±1.1	0.4 ± 0.5	0.2 ± 0.4	35.8 ± 3.6

Note. Differences from the control are significant at *p* ≤ 0.05 (according to the Mann–Whitney test).

**Table 3 ijms-26-07499-t003:** Indices of mitotic phases of meristematic cells of *Allium cepa*, M ± m.

Experimental Group	Mitotic Phase Index, %				Mitotic Phase Index
	PI	MI	AI	TI	
Control	97.89	1.36	0.60	0.12	-
AgNPs@*Artemisia* EtOH/H2O ×40 (70 + 35 H_2_O)	99.60	0. 32	0	0	0.39
AgNPs@*Artemisia* EtOH/EtOAc extract ×45 (73 nm)	99.44	0.39	0.11	0.06	0.56

## Data Availability

The data presented in this study is available on request from the corresponding authors.

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
