# Peer review of "Green Synthesis and Characterization of Silver Nanoparticles Using Artemisia terrae-albae Extracts and Evaluation of Their Cytogenotoxic Effects"

_ijms, 2025, doi:10.3390/ijms26157499_

Round 1
Reviewer 1 Report
Comments and Suggestions for Authors
The goals of the work are 1) to develop a “green” synthetic method of making silver nanoparticles, using molecules extracted from Artemisia terrae-albae as the ligands, and 2) to use test the effects of these nanoparticles on the root growth and mitosis of onions. The goals are reasonable, but the authors need to make a number of changes before the work can be published.
The authors:
- need to carry out more control experiments in their root growth/mitosis experiments before they can attribute the results to the presence of silver nanoparticles and before they can conclude that ”solvent systems play critical role particularly in the modulation of cytogenotoxicity of synthesized silver nanoparticles (AgNPs).” They performed one type of control experiment by placing the onions in water rather than in a silver nanoparticle solution, but they also need to place onions in solutions of silver nitrate and in solutions of Artemisia terrae-albae from the different axtracts, because both silver nitrate and Artemisia terrae-albae are starting materials and could be present in the final solutions and therefore could be causes of the effects on root growth/mitosis.
- need to measure the concentration of silver ions in the nanoparticle solutions using, for example, a silver ion selective electrode. The authors ultrasonicated the silver nanoparticle solutions, saw no change in the UV-visible spectrum of the nanoparticles, and concluded this meant that all of the silver ions had been converted into silver nanoparticles; however, ultrasonication is not a test for conversion to nanoparticles. It is quite possible (and is often the case) in silver nanoparticle syntheses that the solutions still have silver ions in it.
- need to explain more clearly why they have chosen the extract of Artemisia terrae-albae to serve as the ligands for the nanoparticles. They suggest that their synthesis is “greener” than conventional syntheses and could be used in biomedical applications; however, one of the major conventional syntheses is simply to boil a solution of silver nitrate and sodium citrate in water. Sodium citrate comes from citric acid and is not toxic. In fact, this conventional synthesis may be “greener” than what the authors did because it does not involve the use of ethyl alcohol, ethyl acetate, and ammonium hydroxide.
- need to clarify some of the details about their synthetic procedures and subsequent UV-visible analysis so that readers can know the concentrations of all species. Were the masses of the extracts of Artemisia terrae-albae the masses of dried extract? Or were they the masses of some volume of extract solution? In the UV-visible spectra in Figure 1, what do the “x40” and “x60” mean? Were the solutions diluted by these amount before recording the spectra? Or were the spectra multiplied by these amounts? Or do they indicate something else?
The authors should also address a few more points:
- In Section 2.2, presumably the size distribution data were obtained from TEM images, although this is not stated. It would also make sense to place the TEM images in Section 2.3 before the size distribution data that are given in Section 2.2
- In Section 3.4, what are “squash preparations”? This term needs to be defined.
- In Section 3.4, the definitions of the various indices of mitosis should be condensed substantially. The notation of the different mitosis stages (P, M, A, T, N) only needs to be given once, not five times.
- In the second-to-last paragraph of the introduction, the sentence “Although cytogenotoxicity…” is incomplete.
Author Response
Dear Reviewer,
We sincerely thank you for your careful reading of our manuscript and for the valuable comments and suggestions that have significantly contributed to its improvement. We have carefully considered all your questions and suggestions and have endeavoured to answer each of them to the fullest extent possible. The manuscript has been edited according to your recommendations as well as the comments of other reviewers.
Your expertise and constructive critique have helped us make the article more structured, scientifically sound, and clear for readers.
With sincere appreciation and respect,
The Authors
Сomments 1: need to carry out more control experiments in their root growth/mitosis experiments before they can attribute the results to the presence of silver nanoparticles and before they can conclude that ”solvent systems play critical role particularly in the modulation of cytogenotoxicity of synthesized silver nanoparticles (AgNPs).” They performed one type of control experiment by placing the onions in water rather than in a silver nanoparticle solution, but they also need to place onions in solutions of silver nitrate and in solutions of Artemisia terrae-albae from the different axtracts, because both silver nitrate and Artemisia terrae-albae are starting materials and could be present in the final solutions and therefore could be causes of the effects on root growth/mitosis.
Response 1:
Thank you for your valuable comment. We fully agree that the inclusion of additional control groups would significantly enhance the reliability of result interpretation. As part of our planned follow-up study, we intend to incorporate the following control variants:
– treatment of Allium cepa seedlings with a silver nitrate solution at the same concentration used for nanoparticle synthesis;
– exposure to Artemisia terrae-albae extracts obtained using different solvents (ethanol, ethyl acetate, etc.) in the absence of AgNO₃.
This will allow for a more accurate assessment of the contribution of starting components to the observed cytogenetic and physiological effects. In the current study, we focused on comparing nanoparticles synthesized from different solvent fractions as a first step toward a more comprehensive analysis of their mechanism of action. Additional control experiments will be presented in future investigations.
Сomments 2: need to measure the concentration of silver ions in the nanoparticle solutions using, for example, a silver ion selective electrode. The authors ultrasonicated the silver nanoparticle solutions, saw no change in the UV-visible spectrum of the nanoparticles, and concluded this meant that all of the silver ions had been converted into silver nanoparticles; however, ultrasonication is not a test for conversion to nanoparticles. It is quite possible (and is often the case) in silver nanoparticle syntheses that the solutions still have silver ions in it.
Response 2:
We agree that ultrasonic treatment alone is not a sufficient method to confirm the complete conversion of silver ions into nanoparticles. In this study, the absence of changes in the UV-visible spectrum after ultrasonic treatment was used as an indirect indication of colloidal stability, rather than as definitive evidence of full ion reduction.
We appreciate the suggestion to measure the residual concentration of silver ions using a silver ion-selective electrode. In the next stage of our research, we will definitely include this analysis, as it will provide a more accurate characterization of the purity of the synthesized nanoparticles and the extent of silver ion reduction—both of which are critical for interpreting the observed biological effects.
Сomments 3: need to explain more clearly why they have chosen the extract of Artemisia terrae-albae to serve as the ligands for the nanoparticles. They suggest that their synthesis is “greener” than conventional syntheses and could be used in biomedical applications; however, one of the major conventional syntheses is simply to boil a solution of silver nitrate and sodium citrate in water. Sodium citrate comes from citric acid and is not toxic. In fact, this conventional synthesis may be “greener” than what the authors did because it does not involve the use of ethyl alcohol, ethyl acetate, and ammonium hydroxide.
Response 3:
We agree that the synthesis of silver nanoparticles using sodium citrate is indeed a classical, simple, and safe method, which can be considered environmentally friendly.
However, the choice of Artemisia terrae-albae in our study was based on several important factors. First, this plant has a rich phytochemical profile (flavonoids, phenolic compounds, terpenoids, etc.), which allows it to perform a dual role – both reducing and stabilizing silver nanoparticles without the need for synthetic reagents. Second, A. terrae-albae is an endemic species native to the arid regions of Kazakhstan and Central Asia and is traditionally used in folk medicine, making it a promising candidate for locally adapted green technologies.
Moreover, the use of organic solvents such as ethanol and ethyl acetate in this context serves to selectively extract different groups of bioactive compounds, enabling better control over the properties of the resulting nanoparticles. We do not oppose our approach to classical methods but rather offer an alternative pathway, particularly suitable for the development of functionalized AgNPs with tailored properties for biomedical applications.
Сomments 4: need to clarify some of the details about their synthetic procedures and subsequent UV-visible analysis so that readers can know the concentrations of all species. Were the masses of the extracts of Artemisia terrae-albae the masses of dried extract? Or were they the masses of some volume of extract solution? In the UV-visible spectra in Figure 1, what do the “x40” and “x60” mean? Were the solutions diluted by these amount before recording the spectra? Or were the spectra multiplied by these amounts? Or do they indicate something else?
Response 4:
Thank you for the comment and the opportunity to clarify the methodological details.
- In the presented methodology, the indicated masses of Artemisia terrae-albae extracts refer to the mass of the dry residue obtained after solvent evaporation (ethanol or ethyl acetate) under reduced pressure. This dry residue was then used to prepare aqueous solutions for the subsequent synthesis of silver nanoparticles.
- The notations “×40” and “×60” on the UV-visible spectra in Figure 1 indicate the dilution factor of the original extract solution prior to spectral recording (diluted 40 and 60 times, respectively). These notations do not refer to mathematical scaling of the spectra or intensity magnification.
Сomments 5: In Section 2.2, presumably the size distribution data were obtained from TEM images, although this is not stated. It would also make sense to place the TEM images in Section 2.3 before the size distribution data that are given in Section 2.2
Response 5:
Thank you for your valuable observation.
You are correct — the size distribution data presented in Section 2.2 were indeed obtained from the analysis of TEM images using ImageJ software. We acknowledge that this was not clearly stated in the original manuscript and have now added a clarification to Section 2.2 to reflect this.
Additionally, we agree that for improved logical flow and clarity, it would be more appropriate to present the TEM images prior to the size distribution analysis. Accordingly, we have restructured the Results and Discussion section by moving the TEM image descriptions and Figure 3 to precede the histogram data in Section 2.2.
We appreciate your suggestion, which has helped improve the organization and readability of the manuscript.
Сomments 6: In Section 3.4, what are “squash preparations”? This term needs to be defined.
Response 6:
The term “squash preparations” refers to a widely used cytogenetic technique in which meristematic root tips are softened (typically by hydrolysis), stained, and then gently pressed (squashed) between a microscope slide and cover slip to obtain a single-cell layer suitable for microscopic observation. This method allows for clear visualization of chromosomes during different stages of mitosis and is commonly used in plant cytogenetic assays, including the Allium cepa test.
We have added a brief explanation of this term in Section 3.4 to clarify its meaning for the reader.
Сomments 7: In Section 3.4, the definitions of the various indices of mitosis should be condensed substantially. The notation of the different mitosis stages (P, M, A, T, N) only needs to be given once, not five times.
Response 7:
Thank you for your comment.
We agree that repeating the definitions of the mitotic stages (P – prophase, M – metaphase, A – anaphase, T – telophase, N – total number of cells) with each index formula was redundant. As per your suggestion, we have condensed the definitions and provided the notation only once to avoid repetition and improve the clarity of Section 3.4.
We appreciate your recommendation, which helped enhance the logical conciseness and overall presentation of the manuscript.
Сomments 8: In the second-to-last paragraph of the introduction, the sentence “Although cytogenotoxicity…” is incomplete.
Response 8:
Thank you for pointing out this important detail.
We acknowledge that the sentence beginning with “Although cytogenotoxicity...” in the second-to-last paragraph of the Introduction was incomplete and lacked a main clause, which may have caused confusion. The sentence has now been revised for clarity and grammatical correctness. The corrected version provides a complete thought and maintains logical flow within the paragraph.
We appreciate your careful reading, which helped improve the quality and readability of the manuscript.

Reviewer 2 Report
Comments and Suggestions for Authors
In this investigation, green synthesis of AgNPs has been achieved by using plant extracts and also tested for their cytotoxicity. It is very interesting field in the development green chemistry.
The following comments should be addressed for further improvement.
The term "cytogenotoxicity has used frequently (in the title, abstract etc.), actually genotoxicity refers to study of genetic molecules (DNA/RNA/mRNA - genomics). This should be avoided and use it as cytotoxicity.
Artemisia terrae- albae, Allium cepa - binomial should always written with italic style, please check the italic font throughout the MS. Author citation for binomials should be included in the methodology.
Figure 1 - Units of X axis and Y-axis should be typed sharp and clear. Picture quality is not good and grids should be removed.
On what basis 2 different solvents (aqueous ethanol and ethyl acetate) selected? Reasons should be included in the introduction. Have studied negative control experiment (UV-Vis Spec)?
Figure 3- TEM images is not clear (replaced with high quality original picture) and the nanoparticles were not uniformly distributed (much clustered). What is size range of the nanoparticles? how it was measured? Provide scale bar in the TEM picture.
Have you checked the stability of the nanoparticles (DLS)? Without XRD,EDX how it was confirmed?
Author Response
Dear Reviewer,
We sincerely thank you for your careful reading of our manuscript and for the valuable comments and suggestions that have significantly contributed to its improvement. We have carefully considered all your questions and suggestions and have endeavoured to answer each of them to the fullest extent possible. The manuscript has been edited according to your recommendations as well as the comments of other reviewers.
Your expertise and constructive critique have helped us make the article more structured, scientifically sound, and clear for readers.
With sincere appreciation and respect,
The Authors
Comment 1: The term "cytogenotoxicity has used frequently (in the title, abstract etc.), actually genotoxicity refers to study of genetic molecules (DNA/RNA/mRNA - genomics). This should be avoided and use it as cytotoxicity. –
Response 1: We would like to thank you for the clarification of terminology.
Comment 2: Artemisia terrae- albae, Allium cepa - binomial should always written with italic style, please check the italic font throughout the MS. Author citation for binomials should be included in the methodology.
Response 2: We have taken the comments into account and included the authors for the binomials in the methodology.
Comment 3: Figure 1 - Units of X axis and Y-axis should be typed sharp and clear. Picture quality is not good and grids should be removed.
Response 3: This quality of figures was obtained on the equipment. We cannot make it significantly sharper
Comment 4: On what basis 2 different solvents (aqueous ethanol and ethyl acetate) selected? Reasons should be included in the introduction. Have studied negative control experiment (UV-Vis Spec)?
Response 4: What do you mean by “Have studied negative control experiment (UV-Vis Spec)?” We obtained the main extract using ethanol as a solvent and then made the fraction separation after removing the solvent through vacuum evaporation, 80.30 g of the resulting residue was combined and suspended in water. This mixture was then sequentially partitioned using petroleum ether, chloroform, ethyl acetate (EtAc), and butanol to obtain distinct extracts. The process yielded specific extracts as follows: petroleum ether extract (5.2 g), chloroform extract (25.8 g), EtAc extract (16.2 g), and butanol extract (5.4 g).
Comment 5: Figure 3- TEM images is not clear (replaced with high quality original picture) and the nanoparticles were not uniformly distributed (much clustered). What is size range of the nanoparticles? how it was measured? Provide scale bar in the TEM picture.
Response 5: We provided the highest quality level of TEM images that contains the scale bar on the bottom right side of the image. The obtained TEM images were analyzed using ImageJ software, where area measurements were performed to determine the average particle diameter.
Comment 6: Have you checked the stability of the nanoparticles (DLS)? Without XRD,EDX how it was confirmed?
Response 6: The zeta potential and size distribution by DLS was described in previous paper
Dyusebaeva, M. A., Berillo, D. A., Berganayeva, A. E., Berganayeva, G. E., Ibragimova, N. A., Jumabayeva, S. M., ... & Vassilina, G. K. (2023). Antimicrobial activity of silver nanoparticles stabilized by liposoluble extract of Artemisia terrae-albae. Processes, 11(10), 3041. https://doi.org/10.3390/pr11103041
Reviewer 3 Report
Comments and Suggestions for Authors
- In abstract on line 14-15 “AgNPs synthesized from the ethanol extract exhibit notable biostability and high cytotoxic activity” followed by line 16-17 “e significantly more cytotoxic than those observed with the ethyl acetate-derived nanoparticles” then why author want to claim as a non toxic AgNPs?
- The finding line of abstract the author show “findings highlight the potential of green-synthesized AgNPs—especially from ethanol extract—for future biomedical applications” if toxic then why author want to recommend for biomedical aaplication?
- In Introduction the last paragraph also have citation its means everything is from other paper?
- Then whats the rational and novelaty of this study?
- As mentioned in the title “Cytogenotoxic effect” where its is in the results?
- Why ius figure-1 C and D? any relivancy to the study?
- The results section should be re-organized for better readability and coherence
- There is no references at all?
- Author should check grammatical error thoroughly.s
Must be improve
Author Response
Dear Reviewer,
We sincerely thank you for your careful reading of our manuscript and for the valuable comments and suggestions that have significantly contributed to its improvement. We have carefully considered all your questions and suggestions and have endeavoured to answer each of them to the fullest extent possible. The manuscript has been edited according to your recommendations as well as the comments of other reviewers.
Your expertise and constructive critique have helped us make the article more structured, scientifically sound, and clear for readers.
With sincere appreciation and respect,
The Authors
Comments 1: In abstract on line 14-15 “AgNPs synthesized from the ethanol extract exhibit notable biostability and high cytotoxic activity” followed by line 16-17 “e significantly more cytotoxic than those observed with the ethyl acetate-derived nanoparticles” then why author want to claim as a non toxic AgNPs?
Response 1:
In the abstract, we indeed state that AgNPs synthesized using the ethanolic extract of Artemisia terrae-albae exhibit notable biostability and higher cytotoxic activity compared to those derived from the ethyl acetate extract. However, we would like to clarify that we do not claim these AgNPs to be non-toxic. On the contrary, the primary objective of this study was to evaluate their cytogenotoxic potential, not to establish their safety profile.
By "biostability," we refer to the colloidal and structural stability of the nanoparticles over time in suspension, which is important for further biological testing and potential applications. This term does not imply low or no toxicity. We acknowledge that the ethanolic AgNPs demonstrated significant biological effects, including mitotic inhibition and chromosomal aberrations, which point to their cytotoxic and potentially genotoxic properties.
Comments 2: The finding line of abstract the author show “findings highlight the potential of green-synthesized AgNPs—especially from ethanol extract—for future biomedical applications” if toxic then why author want to recommend for biomedical aaplication?
Response 2: Next step of research will be devoted to estimation of difference between cytotoxic effect on normal and cancer cells. This comparative analysis is essential for determining the selectivity and potential therapeutic index of green-synthesized AgNPs in anticancer applications. Relevant studies include the works by
- Juarez-Moreno, K., Gonzalez, E. B., Girón-Vazquez, N., Chávez-Santoscoy, R. A., Mota-Morales, J. D., Perez-Mozqueda, L. L., ... & Bogdanchikova, N. (2017). Comparison of cytotoxicity and genotoxicity effects of silver nanoparticles on human cervix and breast cancer cell lines. Human & experimental toxicology, 36(9), 931-948.
- Yeasmin, S., Datta, H. K., Chaudhuri, S., Malik, D., & Bandyopadhyay, A. (2017). In-vitro anti-cancer activity of shape-controlled silver nanoparticles (AgNPs) in various organ specific cell lines. Journal of Molecular Liquids, 242, 757-766.
- Gurunathan, S., Qasim, M., Park, C., Yoo, H., Kim, J. H., & Hong, K. (2018). Cytotoxic potential and molecular pathway analysis of silver nanoparticles in human colon cancer cells HCT116. International journal of molecular sciences, 19(8), 2269.
Comments 3: In Introduction the last paragraph also have citation its means everything is from other paper?
Response 3: In Introduction the last paragraph should not have reference as we highlight the main focus of our paper.
Comment 4: Then whats the rational and novelaty of this study?
Response 4: The novelty of this study is the cytotoxicity evaluation of stable AgNPs stabilise by Artemisia terrae-albae ethylacetate extracts. As major previous research was focused on the aqueous suspensions of AgNPs.
Comments 5: As mentioned in the title “Cytogenotoxic effect” where its is in the results?
Response 5: Cytogenotoxic effect is ascribed in the section:
2.5. Cell Mitosis studies of EtOH-AgNPs and EtOAc-AgNPs
2.6. Cytogenotoxicity studies of EtOH-AgNPs and EtOAc-AgNPs
Comments 6: Why ius figure-1 C and D? any relivancy to the study?
Response 6: We presented (C) Art aq. extract x60 before ultrasonic treatment; (D) Art aq. extract x60 after ultrasonic treatment as well as (A) before ultrasonic treatment Art H2O x40 + AgNO3 and (B) after ultrasonic treatment Art aq. extract x40 to show difference in size distribution of nanosuspention and therefore surface area and potential difference in reactive oxygen concentration. As one can see from those figures aqueous extract AgNPs are less stable and aggregates over time.
Comments 7: The results section should be re-organized for better readability and coherence
Response 7: We agree with the suggestion and it was addressed appropriately.
Comments 8: There is no references at all?
Response 8: This question is too general could you please clarify it. In the reference list 46 used references presented.
Comments 9: Author should check grammatical error thoroughly.s
Response 9: The manuscript was corrected thoroughly.
Round 2
Reviewer 1 Report
Comments and Suggestions for Authors
The authors made several good changes to the manuscript, but in its present form, the manuscript is still not publishable. I think the authors have two options: 1) carry out the control experiments for this manuscript so that they can make scientific arguments that are supported by solid evidence, or 2) re-write the manuscript and scale back their claims and conclusions significantly so that they claim only that the two solutions have different cytotoxic effects but that they cannot yet determine what the active agent is in the solutions.
Here are my responses to the authors’ responses:
1) In response to my original comment #1 (more control studies need to be carried out before the claim can be made that the AgNPs are responsible for the observed cytotoxic effects), the authors say they will perform these control studies in a future investigation. My comment still stands: the control experiments are needed for the current work. One problem is that for 10 to 15 years, it has been known that in many cases, silver nanoparticle solutions can show cytotoxic effects not because of the nanoparticles themselves but because of the silver ions present in the solution. The ions are present either because of incomplete reduction of the starting reagent or because nanoparticles in essence serve as carriers of silver atoms that are oxidized and then desorb into solution.
2) The silver ion concentrations still need to be measured for the present manuscript, unless the manuscript is completely re-written, as noted above. Also, in the sentence (line 118), “Enhancement of the AgNPs SPR peak…” the portion about indicating the completion of the redox reaction needs to be removed. The lack of change in the UV-visible spectrum indicates the AgNPs are likely stable to the ultrasonication but says nothing about whether the redox reaction was complete.
3) I think that the discussion the authors gave me about why they chose A. terrae-albae and why they use the solvent extractions is good and should be added to manuscript.
4) As in 3), their explanations given to me should be added to the manuscript. Also, the notation that the authors added in Fig. 2 (EtOHx40x60000) is confusing. In the text or in the figure caption, the authors should state more clearly what they mean.
I have a new comment that needs to be addressed before the authors can publish this manuscript:
In my initial review, I neglected to mention a significant issue at the end of the cytotoxicity results section. In the paragraph starting on Line 271, the authors conclude that larger AgNPs (73 nm) appear to be less toxic than smaller AgNPs (40 nm). From their TEM analysis, however, the authors conclude that both of their AgNP samples (formed from EtOH and EtOAc extracts) predominantly have particles that are less than 50 nm in size. Although the authors do not state the average size of the AgNPs, the TEM histograms of both samples indicate that the average sizes are roughly the same (40 nm or slightly lower), but neither sample shows an average anywhere near 73 nm. So, either the authors need to present evidence that they have 73 nm AgNPs, or they need to remove the discussion of the size effects on cytotoxicity.
Finally, while the content of the changes that the authors made, they need to further proofread the changes (e.g., line 100, “cytotoxisity via a knownmethod” should be “cytotoxicity via a known method”).
Author Response
Response to Reviewer Comments
Dear Reviewer,
Thank you for your continued attention to our manuscript. We would like to note that the manuscript has been revised in accordance with your comments, as well as the suggestions of the other reviewers provided after the first round of review.
Below, we provide point-by-point responses to each of your remarks, along with explanations of the corresponding changes made to the manuscript.
We sincerely appreciate your time and valuable input, which have contributed significantly to improving the quality of our work.
With respect and gratitude,
The Authors
Сomments 1: In response to my original comment #1 (more control studies need to be carried out before the claim can be made that the AgNPs are responsible for the observed cytotoxic effects), the authors say they will perform these control studies in a future investigation. My comment still stands: the control experiments are needed for the current work. One problem is that for 10 to 15 years, it has been known that in many cases, silver nanoparticle solutions can show cytotoxic effects not because of the nanoparticles themselves but because of the silver ions present in the solution. The ions are present either because of incomplete reduction of the starting reagent or because nanoparticles in essence serve as carriers of silver atoms that are oxidized and then desorb into solution.
Response 1: We agree with your point of view. We did the control experiment for presence of silver ions in solution after the AgNPs formation. The reaction with sodium chloride did not result white precipitate formation as silver chloride, moreover additional addition of glucose solution did not show increase of SPR peak of AgNPs. Indicating again on absence of free silver ions.
Сomments 2: The silver ion concentrations still need to be measured for the present manuscript, unless the manuscript is completely re-written, as noted above. Also, in the sentence (line 118), “Enhancement of the AgNPs SPR peak…” the portion about indicating the completion of the redox reaction needs to be removed. The lack of change in the UV-visible spectrum indicates the AgNPs are likely stable to the ultrasonication but says nothing about whether the redox reaction was complete.
Response 2: We agree with your point . We corrected the text suggested. The lack of change in the UV-visible spectrum of AgNPs after ultrasonication may be related to good stability of AgNPs. There are many publications stated that ultrasonication induces AgNPs formation. https://scholar.google.com/scholar?hl=ru&as_sdt=0%2C5&authuser=1&q=ultrasonication+induce+AgNPs+formation&btnG=
We checked presence of silver ions in solution after the AgNPs formation. The reaction with sodium chloride did not result white precipitate formation as silver chloride, moreover additional addition of glucose solution did not show increase of SPR peak of AgNPs. Indicating again on absence of free silver ions.
Сomments 3: I think that the discussion the authors gave me about why they chose A. terrae-albae and why they use the solvent extractions is good and should be added to manuscript.
Response 3: We have added a justification for the selection of A. terrae-albae in the Introduction and also expanded the Discussion section.
The following text was added to section 2.4:
Recently we utilized solvent solvent extractions method. A. terrae‑albae was selected for its exceptionally high α‑santonin content, and solvent extraction was employed to efficiently isolate its bioactive constituents. In extract 3, α‑santonin made up 66.33% of the yield (RT 18.2 min), alongside minor amounts of 2,3‑dihydro‑4H‑pyran‑4‑one (3.2%), 3,5,5‑trimethylcyclohexylisophosphofloride (6.73%), lumisantonine (3.74%), and 6‑nitro‑2‑phenyl‑4‑quinolinol (8.0%). Extract 4 contained 39.39% α‑santonin, 10.05% butanoic acid, and 3.28% anhydro‑β‑retinol, while extract 7 mirrored extract 3 with 66.35% α‑santonin, plus 4.9% 4‑H‑pyran‑4‑one‑2,3, 5.9% 7‑ethyl‑4‑nonanone, and 2.58% lumisantonine. Since antiviral activity ranked No 4 > No 3 = No 7 and tracked with α‑santonin levels, these results implicate santonin as the primary active component [32].
Сomments 4: As in 3), their explanations given to me should be added to the manuscript. Also, the notation that the authors added in Fig. 2 (EtOHx40x60000) is confusing. In the text or in the figure caption, the authors should state more clearly what they mean.
Response 4: Thank you for the comment. We agree that the original notation in Figure 2 (now Figure 3) may have been unclear. To avoid any misunderstanding, we have revised the figure caption and clarified the notation in the main text. Specifically, “EtOH×40×60000” now refers to AgNPs synthesized using Artemisia ethanol extract diluted 40 times with water and imaged at 60000× magnification. The updated figure caption now provides a clearer and more detailed explanation.
Сomments 5: I have a new comment that needs to be addressed before the authors can publish this manuscript:
In my initial review, I neglected to mention a significant issue at the end of the cytotoxicity results section. In the paragraph starting on Line 271, the authors conclude that larger AgNPs (73 nm) appear to be less toxic than smaller AgNPs (40 nm). From their TEM analysis, however, the authors conclude that both of their AgNP samples (formed from EtOH and EtOAc extracts) predominantly have particles that are less than 50 nm in size. Although the authors do not state the average size of the AgNPs, the TEM histograms of both samples indicate that the average sizes are roughly the same (40 nm or slightly lower), but neither sample shows an average anywhere near 73 nm. So, either the authors need to present evidence that they have 73 nm AgNPs, or they need to remove the discussion of the size effects on cytotoxicity.
Response 5:
We sincerely thank the reviewer for this important remark. After a thorough revision, we agree that the reference to AgNPs of “40 nm” and “73 nm” in the cytotoxicity section was inaccurate, as it does not correspond to the TEM data, which show that the majority of nanoparticles in both samples (based on EtOH and EtOAc extracts) are smaller than 50 nm.
This discrepancy resulted from a misstatement in the previous version of the text. In the revised manuscript, we have corrected the relevant paragraph and removed the specific numerical size values that were not supported by TEM analysis. Instead, the discussion now focuses on relative differences in particle morphology and size distribution based on TEM results and histograms (Figure 4). The sample synthesized using the EtOH extract (diluted ×40) showed a more uniform and smaller size distribution, whereas the EtOAc extract sample (diluted ×45) exhibited a broader and more heterogeneous distribution with a greater proportion of larger particles.
We clarified in the revised text (Lines 311-330) that the observed differences in cytotoxicity are interpreted in relation to the differences in size distribution and aggregation tendency, rather than based on absolute particle sizes.
Once again, we sincerely thank you for your attention to detail and for helping us improve the accuracy and clarity of our manuscript.
Сomments 6: Finally, while the content of the changes that the authors made, they need to further proofread the changes (e.g., line 100, “cytotoxisity via a knownmethod” should be “cytotoxicity via a known method”).
Response 6:
We agree with the comment, it was a mistake, that was corrected. One can obsertve quite a few large particles with a size exeeds 100 and 150 nm. These particles has tendency to precipitate for a short period of time, therefore less likely can be detected using light sacttering method. Histogram 4B is sparser and biased towards larger particles. Both distributions are asymmetrical. Most likely the extract has fewer compounds that has ability to rapidly stabilize growing AgNPs, that leads to growth of large perticles. The larger the particle less ratio of surface to size and in turn less activity.

Reviewer 2 Report
Comments and Suggestions for Authors
The manuscript still need to be revised based on the following comments
Response for the earlier comments not properly address the issues. for example
comment 3: Figure 1. X axis units are not uniform and not visible as in Fig.1B.
Comment 4: I am asking about the control experiments (UV-Vis spectra for silver nitrate solution and plant extract )
Answer to comment no.6: Author stated that the stability test for Silver nanoparticle was done and it is already published (Dyusebaeva, M. A., Berillo, D. A., Berganayeva, A. E., Berganayeva, G. E., Ibragimova, N. A., Jumabayeva, S. M., ... & Vassilina, G. K. (2023). Antimicrobial activity of silver nanoparticles stabilized by liposoluble extract of Artemisia terrae-albae. Processes, 11(10), 3041. https://doi.org/10.3390/pr11103041)
What is purpose of using the same plant and similar extraction methods/similar solvents/UV-Vis data/electron micrograph has been used in this manuscript? Characterization of silver nanoparticles were already published earlier, why it is included in this MS? How it is different from the previously published one?
Author Response
Response to Reviewer Comments
Dear Reviewer,
Thank you for your continued attention to our manuscript. We would like to note that the manuscript has been revised in accordance with your comments, as well as the suggestions of the other reviewers provided after the first round of review.
Below, we provide point-by-point responses to each of your remarks, along with explanations of the corresponding changes made to the manuscript.
We sincerely appreciate your time and valuable input, which have contributed significantly to improving the quality of our work.
With respect and gratitude,
The Authors
Сomment 1: Figure 1. X axis units are not uniform and not visible as in Fig.1B.
Response 1: Figure 1. was substituted for more suitable.
Comment 2: I am asking about the control experiments (UV-Vis spectra for silver nitrate solution and plant extract )
Response 2: The control experiment spectrum was added to the figure 1. Silver nitrate is the transparent solution that almost can not be seen in UV vis before 300nm at utilized concentration.
Comment 3: Answer to comment no.6: Author stated that the stability test for Silver nanoparticle was done and it is already published (Dyusebaeva, M. A., Berillo, D. A., Berganayeva, A. E., Berganayeva, G. E., Ibragimova, N. A., Jumabayeva, S. M., ... & Vassilina, G. K. (2023). Antimicrobial activity of silver nanoparticles stabilized by liposoluble extract of Artemisia terrae-albae. Processes, 11(10), 3041. https://doi.org/10.3390/pr11103041)
What is purpose of using the same plant and similar extraction methods/similar solvents/UV-Vis data/electron micrograph has been used in this manuscript? Characterization of silver nanoparticles were already published earlier, why it is included in this MS? How it is different from the previously published one?
Response 3: This manuscript devoted to evaluation of cytotoxisity of AgNPs@Artemisia EtOH/H2O x40 dilution of initial plant extract. Herein, we report the histological assessment of AgNPs cytotoxi`sity via a known method modified Allium cepa assay, which involved the pre-treatment of bulbs at 4 °C for 72 hours to synchronize the mitotic cycle in root meristem cells.
This experiment was done later therefore it was not included in previous paper. We tried to improve synthesis method of AgNPs using various conditions and parts of the extract, however obtained nanoparticles were not as stable as AgNPs@Artemisia EtOH/H2O x40, therefre we did continue to work with it.
Reviewer 3 Report
Comments and Suggestions for Authors
Author address all the comments, i will recomend for publication
Author Response
Dear Reviewer,
Thank you for your positive evaluation of our work and for the valuable comments that helped improve the manuscript. We sincerely appreciate your attention, kind approach, and your decision to recommend our article for publication.
We have also made additional changes to the manuscript in accordance with the comments and suggestions of the other reviewers (highlighted in yellow).
With respect and gratitude,
The Authors
Round 3
Reviewer 1 Report
Comments and Suggestions for Authors
The authors have again made a number of improvements, but they need to better proofread their manuscript and make sure that the spelling is correct, figure numbers are correct in the body of the text (especially since the figure numbers have been changed since the last revision), and that the language and notation are consistent throughout the manuscript (e.g., in a simple example, they should always use AgNPs@Art or AgNPs@Artemisia, but not both).
Here are a few more specific comments:
Line 124: I don’t think a period goes after the word “Berg” since Berg is the complete name of the person. Putting in the period makes “raw material” look like an incomplete sentence and interrupts the flow of logic.
Line 128: change “In this study we show simple method” to “In the present study, we show a simple method”. You used the word “this” in the previous sentence to refer to previous work, so using “this” again is a little confusing.
Line 132: remove the word “diluted”. Isn’t the left sample the extract that is not yet diluted? And isn’t the extract just in EtOH, not EtOH/H2O? So shouldn’t the caption of Fig 1A be “Artemisia EtOH extract before the synthesis…”?
Lines 133 and 134: change “initian” to “initial”
Line 134: change “x450 x600 and x4500” to “x450, x600, and x4500”
Line 137: The meaning of the sentence is not clear. Do you mean: “Use of the Artemisia extract that is diluted 40 times in water is preferable than using one diluted 60 times in water, perhaps because it contains a higher concentration of compounds valuable for AgNP stabilization at the seeding stage.”?
Line 146: change “of AgNPs. Indicating again on absence of free silver ions.” to “of AgNPs, indicating again an absence of free silver ions.”
Figure 2: There are several confusing items that need to be fixed. The caption for panels C and D indicate the nanoparticles are made from the EtOH extract, but the additional explanation and the inset in panel C indicate the nanoparticles are made from the EtOAc extract. Also, in Figure 2A, the caption suggests that you added additional AgNO3, but the spectrum is just for the nanoparticles made from the EtOH extract diluted 40x. Right?
Figure 2C: Although the authors do not show a picture of the original EtOAc extract, the green spectrum in Figure 2C shows almost no absorption in the visible, implying that the original extract should be almost colorless. Is that really correct?
Line 163: this paragraph is not part of the figure caption, so it belongs in the text and should use complete sentences.
Line 230 (Section 2.4): The authors added a long section (almost two pages) summarizing their results from a previous study, but they need to explain how this is directly relevant to the current study that uses AgNPs. Also, once the authors show its relevance, I suspect that the added section could be shortened significantly.
Lines 335-356: As in the last comment, the authors need to show how this new section is directly relevant to the current study. They need to explicitly provide a conceptual link between the prior work and the current work.
Author Response
Dear Reviewer,
We would like to express our sincere gratitude for your time and valuable comments, which have significantly contributed to improving the quality of our work. The manuscript has been revised in accordance with your suggestions, as well as the feedback received following the third round of review.
Below, we provide point-by-point responses to each of your comments, along with explanations of the corresponding changes made to the manuscript.
With respect and appreciation,
The Authors
Comment 1: Line 124: I don’t think a period goes after the word “Berg” since Berg is the complete name of the person. Putting in the period makes “raw material” look like an incomplete sentence and interrupts the flow of logic.
Response 1: We agree with comment, typo was corrected.
Comment 2: Line 128: change “In this study we show simple method” to “In the present study, we show a simple method”. You used the word “this” in the previous sentence to refer to previous work, so using “this” again is a little confusing.
Response 2: We agree with comment and accept your suggestion of text correction. Thanks.
Comment 3: Line 132: remove the word “diluted”. Isn’t the left sample the extract that is not yet diluted? And isn’t the extract just in EtOH, not EtOH/H2O? So shouldn’t the caption of Fig 1A be “Artemisia EtOH extract before the synthesis…”?
Response 3: Corrected according to the above comment. The initial extract of Artemisia was obtained using extraction with ethanol, evaporated diluted with water and additional fractional extraction with ethylacetate was performed. Then extract was evaporated and dissolved in water for the following AgNPs synthesis therefore we name it as Artemisia EtOH/H2O/EtAc
before ultrasonic treatment AgNPs@Artemisia EtOH/H2O/EtAc dilutedx40 + AgNO3 and (B) after ultrasonic treatment Artemisia EtOH/H2O diluted x40; (C) AgNPs@Artemisia EtOH/H2O/EtAc extract diluted x60 water before ultrasonic treatment and then additionally 10 times diluted with water and the control sample Artemisia EtOH/H2O/EtAc extract; (D) Artemisia EtOH/H2O/EtAc extract diluted x60 water after ultrasonic treatment.
Comment 4: Lines 133 and 134: change “initian” to “initial”
Response 4: We agree with comment, typo was corrected.
Comment 5: Line 134: change “x450 x600 and x4500” to “x450, x600, and x4500”
Response 5: We agree with the comment that the confused expression was used, and it was corrected.
- B) AgNPs@Artemisia EtOH/H2O x45 and AgNPs@Artemisia EtOH/H2O x60 dilution of initial plant extract for the synthesis of particles and following dilution to 10 and 100 times respectively.
Comment 6: Line 137: The meaning of the sentence is not clear. Do you mean: “Use of the Artemisia extract that is diluted 40 times in water is preferable than using one diluted 60 times in water, perhaps because it contains a higher concentration of compounds valuable for AgNP stabilization at the seeding stage.”?
Response 6: Yes, we meant to say this. We agree with confused construction of the sentence and accept your suggestion. Thanks.
Comment 7: Line 146: change “of AgNPs. Indicating again on absence of free silver ions.” to “of AgNPs, indicating again an absence of free silver ions.”
Response 7: We agree with comment, typo was corrected.
Comment 8: Figure 2: There are several confusing items that need to be fixed. The caption for panels C and D indicate the nanoparticles are made from the EtOH extract, but the additional explanation and the inset in panel C indicate the nanoparticles are made from the EtOAc extract. Also, in Figure 2A, the caption suggests that you added additional AgNO3, but the spectrum is just for the nanoparticles made from the EtOH extract diluted 40x. Right?
Response 8: We agree with numerous inaccuracies, that were corrected.
Comment 9: Figure 2C: Although the authors do not show a picture of the original EtOAc extract, the green spectrum in Figure 2C shows almost no absorption in the visible, implying that the original extract should be almost colorless. Is that really correct?
Response 9: A solution may appear colored not due to absorption, but due to light emission — fluorescence. For example, a substance may absorb in the UV region and emit light in the visible range. This creates the impression of color, even though the absorption spectrum in the visible region does not show pronounced bands. Examples of such substances include coumarin, quinine, and chlorophylls — they can fluoresce in the blue or red region of the spectrum.For example, the study demonstrates how mixtures of plant extracts (such as anthocyanins from pomegranate and curcumin) emit visible light when excited by UV light (380 nm): blue (~440 nm), green (~522 nm), and red (~590 nm). At the same time, the visible absorption spectrum may not show pronounced bands. https://pmc.ncbi.nlm.nih.gov/articles/PMC4470329/
Here are some more examples:
- https://www.researchgate.net/figure/UV-Visible-spectra-black-and-fluorescence-spectra-a-Pom-extract-blue-line-where_fig8_278792919;
- https://www.mdpi.com/1999-4907/14/6/1094
Other similar extracts for instance Artemisia annua leaf extract does not have any significant peak in visible region of the spectra. UV-Visble spectra of Artemisia annua leaf extract and biosynthesized... | Download Scientific Diagram
Comment 10: Line 163: this paragraph is not part of the figure caption, so it belongs in the text and should use complete sentences.
Response 10: We agree with the comment. The sentence was removed.
Comment 11: Line 230 (Section 2.4): The authors added a long section (almost two pages) summarizing their results from a previous study, but they need to explain how this is directly relevant to the current study that uses AgNPs. Also, once the authors show its relevance, I suspect that the added section could be shortened significantly.
Response 11: Thank you for your valuable comment. We agree that the previously added section summarizing earlier research required clarification of its relevance to the current study.
We have previously investigated the phytochemical composition and biological activity of Artemisia terrae-albae collected from Kazakhstan. For instance, one of our prior studies focused on the qualitative and quantitative phytochemical analysis of this medicinal plant, where we evaluated the antimicrobial activity of both crude extracts and silver nanoparticles (AgNPs) stabilized by Artemisia terrae-albae extracts. The AgNPs synthesized from these extracts exhibited average particle sizes of ~82 nm with a negative surface charge, and displayed antimicrobial activity against Pseudomonas aeruginosa, Staphylococcus aureus, Escherichia coli, and Candida albicans. These findings provided a basis for exploring plant-mediated AgNPs as potential antimicrobial agents. https://doi.org/10.3390/pr11103041
In another study, we investigated the biologically active compounds of Artemisia cina Berg and demonstrated its antiviral activity against SARS-CoV-2 for the first time. Moreover, we conducted comprehensive toxicity assessments, including acute toxicity (LD₅₀ > 2 g/kg) and biochemical urine analysis in mice, showing no significant organ toxicity. https://doi.org/10.3390/molecules28145413
Based on this previous experience with Artemisia species and their bioactive extracts, the current study advances this work by investigating the cytotoxic properties of AgNPs synthesized via green chemistry approaches using aqueous-ethanol and ethyl acetate extracts of A. terrae-albae. Unlike our earlier antimicrobial-focused studies, here we apply a modified Allium cepa assay to assess mitotic activity and chromosomal effects, correlating cytotoxicity with particle size and morphology.
Thus, we believe that the inclusion of the previous research is justified, as it establishes the rationale for selecting Artemisia terrae-albae and informs our methodological approach in the current cytotoxicity assessment.
Comment 12: Lines 335-356: As in the last comment, the authors need to show how this new section is directly relevant to the current study. They need to explicitly provide a conceptual link between the prior work and the current work.
Response 12: We have revised the section accordingly to clarify its relevance to our current study. The cited studies on the genotoxic and cytotoxic effects of various Artemisia species provide important background supporting our choice of model and methodology. Specifically, these prior findings highlight the potential of Artemisia-based products to affect cell division and genetic stability in both normal and cancer cells. This is directly relevant to our current investigation, where we evaluate the impact of AgNPs@Artemisia extracts on mitosis in Allium cepa meristematic cells. The conceptual link lies in the shared focus on cell division inhibition and DNA damage potential, which together support the rationale for applying Allium test systems to assess mitotic activity and potential genotoxicity of our test samples.

Reviewer 2 Report
Comments and Suggestions for Authors
Accept in Present form
Author Response

(The authors gave the same response as above.)
